# Curricular Innovation for Food Security

Irma Cecilia Castillo Escalante [1] and Adán Guillermo Ramírez García [2,*]

1    Centro de Bachillerato Tecnológico Agropecuario No.197, Ciudad Obregón 85203, Mexico
2    Centro Regional Universitario del Noroeste, Universidad Autónoma Chapingo, Texcoco 56230, Mexico
*    Correspondence: gramirezg@taurus.chapingo.mx

**Abstract:** The curriculum is a dynamic and continuous process that must be adapted to the requirements and needs of specific contexts to respond to current problems. Currently, it is estimated that more than 828 million people in the world suffer from hunger, so food security must be based on the production and availability of food and physical and economic access to food, as well as its safety. For the above, this research aims to propose a curricular innovation to the professional competences of the study program of the Agricultural Technician (TAP) that is offered in the Agricultural Technological Baccalaureate Centers to train technicians who contribute to food security in the region. A methodology with a mixed perspective with a qualitative approach and descriptive exploratory character was used. Based on the primary information and data collection, it was possible to specify and substantiate a curricular proposal according to the training needs of specialized technicians, who develop professional skills to promote food security, the establishment of agricultural production system, management of livestock production systems, operation of agro-industrial production system, and the design and execution of projects to promote the cultivation of food from small-scale families.

**Keywords:** education; interdisciplinarity; food security; transdiscipline; innovation; curriculum





## 1. Introduction

An educational innovation implies the implementation of a significant change in the teaching–learning process, which incorporates methodologies that modify and raise the quality of teaching in all its areas [1]. Any initiative of this type must focus on the student; that is, more and more thought is given to helping the learner develop their own criteria and that concepts and ideas generate social and formative awareness for work and change in society [2].

Innovation proposes new critical activities and new pedagogical procedures involving a differentiated management, transforming it into a vehicle that allows us to observe the needs and dimension of the innovation that is intended to be carried out. Institutions take the option of introducing these changes, which turn out to be novel with the intention of optimizing the administration of educational programs [3].

For [4], the constructivist educational model aims to promote analysis, criticality, and reflection. The skill focuses more on finding strategies and resources that facilitate the inter-learning process, where the student is the main character and the teacher becomes an academic guide. In this model, planning is active, it is modified according to the different moments of the chair, and the student's participation is validated according to their reflections and readings, so they develop cognitive, motor, artistic, and socio-affective skills to gain reflective autonomy and metacognition in their critical thinking. The constructivist educational model gives way to authentic evaluation, focused on student participation, valuing their significant learning situations and considering the contextualized knowledge applied in the model [5].

The educational environment aims to improve the quality of the educational system, promoting an integral development of professionals, achieving the active participation of students in academic work, and satisfying the needs of an increasingly specialized

society [6]. In this sense, the study of [7] can be mentioned, who carried out a curricular design based on competencies; the said design was approached from a qualitative socio-formative approach. It was carried out in four phases:

1. Identification of problems, which can have practical solutions.
2. Diagnostic reflection, to understand and analyze the problem.
3. Planning, through SWOT analysis, in order to guide the work of educational value.
4. Action—Observation: A record and conclusion is made.

The new approaches to curricular innovations have as their guiding axis the concept of complexity; this concept indicates that it is necessary to learn to apprehend thinking that the world system is complex and dynamic, and the vision of educated subjects must be oriented from a transdisciplinary, critical and systemic perspective [8–10].

Curricular innovation seeks that those interested in continuing their educational process have tools and means to access knowledge that helps them change their lives, in a positive way, and at the same time respect their ideals and beliefs. Although, in this globalized world, with the environment pressing, culture and traditions are at risk of being lost, and more if education is planned from a center and does not consider the peripheries and minorities [11].

Thus, the vision of complexity invites curricular innovation to not give in to the pressures of large industries and world economies, but that educational plans and programs guide the efforts of local and regional society to solve global problems through local actions [12–14].

For several years, the educational process has been in crisis. A pedagogical–didactic crisis that one does not want to recognize. Teaching focused on the teacher's explanation in front of a group of students is still the predominant model of classroom work, with curricula full of information about everything a professional is supposed to know. An encyclopedic vision has permeated these plans, and only work during the COVID-19 pandemic forced the need on some occasions to think about what the substantive topics are that a student needs to know and, above all, what are the intellectual processes that need to be developed to be a professional for the XXI century [15].

The need for an urgent change in our way of being, knowing, and being on the planet is becoming increasingly evident, which is reflected in the increase in the number of careers that include sustainability in their curriculum. However, most are curricula structured to continue responding to the dynamics of economic accumulation and around a single discipline, which makes it impossible to grasp the complexity that characterizes the ecological and socio-economic crisis [16].

With regard to food security, according to the Food and Agriculture Organization of the United Nations [17], more than 700 million people worldwide currently suffer from hunger. This number increased during the COVID-19 pandemic. Between 2019 and 2020, approximately 103 million people were affected, and 46 million more people in 2021. According to their geographical distribution, hunger affects 278 million people in Africa (20.2% of the population), 425 million in Asia (9.1% of the population), and 56.5 million in Latin America (8.6% of the population).

In terms of food security, according to [18], it is achieved when, at the individual, household, national, and global level, everyone at all times has physical and economic access to sufficient safe and nutritious food to meet their dietary needs and food preferences, in order to lead an active and healthy life. For its study, it is approached from four dimensions: availability, physical and economic access, use and stability, and maintenance of food. The reason why any modification to one of them could cause conditions of food insecurity is discussed in [19].

For its part, [20] ensures that food security is based on the recognition of the right to food, as a fundamental human right, and that it includes the availability of food obtained from family production units and/or from the communities themselves; access to food not directly produced, which implies the availability of resources for its acquisition; resources obtained through the realization of other activities by the different members of the pro-

ductive units; and the good use of the food produced or obtained, always according to the culture and traditions of the families and communities.

In Mexico, family farming continues to be one of the most widespread and dynamic sectors in rural areas, but its importance has not been sufficiently recognized or valued in national public policies. It has not been given the role that could be given to it, in terms of the diversity it represents and in recognition of the contribution it makes to the values of production and conservation of plant and animal genetic resources [21].

It is rural families, like no other human population, that depend most on environmental resources and services, whose ways of life are closely related to the context around them, a situation that makes them especially valuable as guardians of natural resources and particularly those vulnerable to environmental degradation. Relying so decisively on these resources, they developed sustainable management techniques called traditional agricultural technology [22].

In addition to its importance as a supplier of food and contributing to food security, family farming contributes to the development of rural territories and communities because it is a generator of employment and a source of income; it is a productive model that favors the attachment of the family to the rural environment; it is a socio-economic sector that has the potential to create poles of economic development and marketing networks; it favors the preservation of endogenous plant and animal species in the region; and it generates direct and indirect employment, to the extent that its activities are market-oriented and incorporate value into products before they are marketed and preserves and enhances cultural aspects, skills, abilities, and traditions [23].

In this sense, according to [24], public policies distinguish different levels of food security, giving priority to economic and health policies. However, it is necessary to consider an educational policy that promotes training through formal and non-formal education of technicians and, where appropriate, graduates and engineers, to combat food poverty and contribute to food security.

For the above, this research aims to propose a curricular innovation to the professional competences of the study program of the Agricultural Technician (TAP) that is offered in the Agricultural Technological Baccalaureate Centers to train technicians who contribute to the food security of the region.

## 2. Materials and Methods

The Centro de Bachillerato Tecnológico Agropecuario (CBTA) are public schools of upper secondary education that offer technological baccalaureate programs with a focus on agriculture, livestock, fishing, and agribusiness. At the end of the studies and complying with the requirements established in the school control manual, graduates can obtain a degree and professional license of the career that was studied.

They are distributed throughout the territory of the United Mexican States and are administered by the Federal Government through the Ministry of Public Education. Currently, the educational offer consists of 31 careers taught in 335 schools, and education is offered to more than 170 thousand students [25,26].

Regarding the offer of a degree in food safety, there are several options, which are listed below:

1. University of Navarra (Spain)
2. University of Almería (Spain)
3. University of Córdoba (Spain)
4. University of Granada (Spain)
5. University of Las Palmas de Gran Canaria (Spain)
6. University of Lleida (Spain)
7. University of Zaragoza (Spain)
8. University of Buenos Aires (Argentina)
9. National University of La Plata (Argentina)
10. National University of Tucumán (Argentina)

11.   Open and Distance University of Mexico
12.   Universidad Autónoma de Nuevo León (Mexico)
13.   Universidad del Valle (Colombia)
14.   National University of Colombia (Colombia)
15.   National Agrarian University La Molina (Peru)
16.   National University of Trujillo (Peru)
17.   University of San Carlos de Guatemala (Guatemala)
18.   National Agrarian University (Nicaragua)
19.   National University of Asunción (Paraguay)
20.   Università degli Studi di Milano (Italy)

However, in an exhaustive review in different databases, it was not possible to find the training of food safety technicians at the Baccalaureate level, hence the relevance of this proposal, which seeks to encourage graduates to join the primary sector when graduating with the title of agricultural technician in food security, and where appropriate continue their university studies in topics related to environmental, agricultural, forestry, and/or rural development sciences.

The research methodology used is mixed, and is an alternative to address topics concerning the educational field, considers both the qualitative and quantitative approach, and the complexity of this type of studies, which throughout the research are inseparable [27].

Its character is exploratory, because the research topic is little studied and is a novel topic. Its logic is descriptive and seeks to specify the differential properties of people, groups, communities, or any other phenomenon that is subject to analysis [28].

With this research, we want to systematize educational practice through curricular innovation. Systematization is understood as the process of monitoring, and the systemic recording of experiences, which allows one to discover, explain, and interpret the logic of the process studied and its causes. This allows us to build and contribute to a system of theoretical and practical knowledge that can be contrasted and replicated with other experiences [29].

To elaborate the curricular design proposal of the Agricultural Technician (TAP), sustainable development and food security, together with curricular innovation through competencies and under the constructivist approach, was used as a theoretical reference [30–32].

Data collection was carried out from workshops; semi-structured interviews; surveys of managers, teachers, and students of the campus under study; and rescue expert opinions. The instruments used during the methodological process followed in the research are described below.

*2.1. Workshops*

Three workshops were held: diagnosis, proposal, and validation, in full with the Academy of Agricultural Technician (TAP), which is made up of six professors who are the ones who teach most of the courses of the Agricultural Technician career. The three workshops were held with school authorities, members of the Academy of Agricultural Technician, experts in the field, and graduates. From the authorities of the campus, the Director, the Academic Sub-directorate, the Academic Coordinator, and some members of the Technical Council participated.

The diagnostic workshop aimed to review the graduation profile, assess the interest in the profile, and build the state-of-the-art of the subject to be investigated, through participatory workshops and employing work dynamics in order to specify the critical points and strengths of the current curriculum. The proposal workshop aimed to analyze the progress of the curricular innovation proposal through the systematization of the information generated in the diagnostic workshop in order to specify the final proposal. The validation workshop aimed to present the curricular innovation proposal in order to give it the go-ahead and, where appropriate, present it to the competent authorities.

*2.2. Semi-Structured Interviews*

Sixteen professionals and experts on educational innovation issues were interviewed; on food security. The questions included closed, multiple-choice, and open questions, where they could freely express their opinion on the subjects in question.

The purpose of the interview with professionals of the curriculum was to deepen the perception of the technical careers of the Agricultural Technological Baccalaureate Center (CBTA 197) and its relationship with respect to food security, the perception of the contents of the curriculum of the Agricultural Technician, the order and organization of said contents, the teaching–learning processes, professional competences, generic and specific competences of the professional component, and insertion sites and their needs.

*2.3. Surveys*

The survey was conducted among CBTA 197 students in the sixth semester, through the Google Applications platform using the Forms tool and distributed via WhatsApp. The total population of the sixth semester of the campus is 500 students, of which 250 belong to the career of Agricultural Technician.

Sample size was calculated with simple random sampling for proportions, with 95% reliability and 5% accuracy or sampling error. The following formula was used:

$$n = \frac{NZ^2 \propto /2\, Pn\, qn}{N\, d^2 + Z^2 \propto /2\, Pnqn}$$

where:

$N$ = Population size
$d$ = Precision
$Z\, \alpha/2$ = Reliability
$Pn$ = Proportion with the characteristic of interest
$qn$ = Proportion without the characteristic of interest

From a population of 250 students in the sixth semester of the Agricultural Technical Career (TAP), a questionnaire was applied to a representative sample, which was calculated as follows:

$N$ = 250
$d$ = 0.75
$Z\, \alpha/2$ = 1.64
$Pn$ = 0.25
$qn$ = 0.25

The sample size was calculated with the assumption of maximum variance: $p = 0.25$ and $q = 0.25$. The sample size was $n = 87.5$ students.

With this sample size defined, the survey was conducted by randomly selecting the students who were interviewed. For the systematization of the answers, these were recorded and processed in a Microsoft Excel 2010 spreadsheet developed by Microsoft for Windows.

The data collection process of the sixth semester students of the Agricultural Technician (TAP) was applied to 88 students, where 53% were women and 47% were men, between 17 and 19 years old. Figure 1 describes the age and gender of the students surveyed from CBTA 197; in the case of the second semester it applied to 90% of the population, and in the case of the sixth semester it was adjusted to the size of the statistical sample estimated at 88 students.

Eighty-eight surveys of the total population of sixth semester students were applied, and the professional skills of their preference were selected. The technical career is studied in five modules and is included from the second to the sixth semester. Of the respondents, 44 were women (50%) and 44 were men (50%). As for the ages of the students surveyed, 77% were 18 years old, 16% were 17 years old, and the remaining 7% were 19 years old.

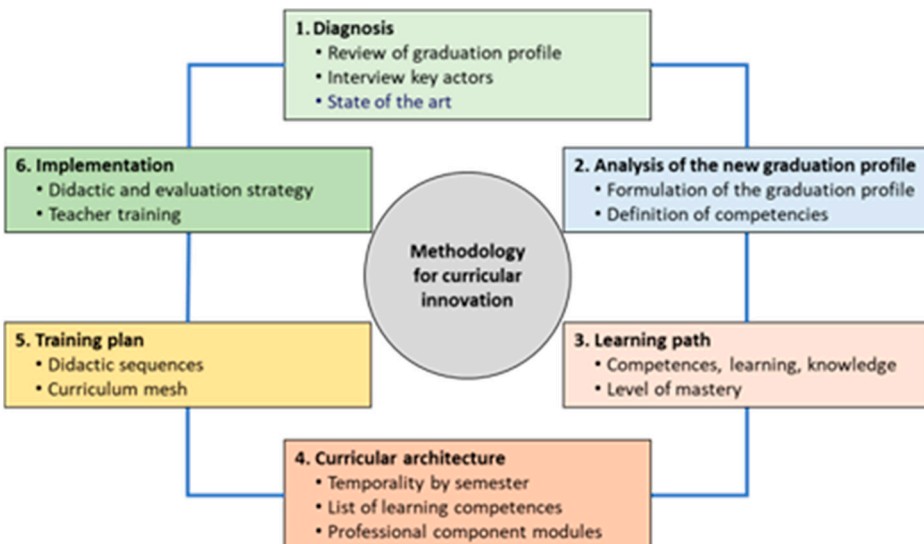

**Figure 1.** Methodological procedure Curricular innovation. Source: Adapted from [3].

### 2.4. Inclusion and Exclusion Criteria

In the selection, it was considered that the interviewees were directly related to the agricultural technician, agricultural production, and the delivery of the agricultural professional component: graduates and students of CBTAS of the agricultural technical career. During the selection process, those students who were "not" CBTAS students, those who "did not" belong to a CBTAS specialty and finally those who were "not" related to the agricultural sector were excluded.

### 2.5. Methodological Procedure for Curricular Innovation

The main purpose of any curricular innovation process is to renew the quality, translated into the effectiveness of teaching–learning and the relevance that refers to the usefulness for society of the degrees, certificates, or titles that are granted in formal education. For the present research, with regard to curricular innovation, the proposal of [3] is taken up. Among the characteristics that support its proposal, is to associate the orientation of the training process that encompasses the knowledge of teachers, managers, and experts, such as the importance of change as an opportunity to improve through generic and specific competences that have coherence with the management procedures for the achievement of knowledge, learning, and competences, as well as the educational quality that the present reality demands. The methodological procedure for the curricular innovation proposal is shown in Figure 1.

### 2.6. Data Analysis

For the data analysis, the information was entered into the Excel program of the Microsoft office 2017 software, where the data of averages, the percentages of both teachers and students, were provided. On the other hand, in the interview stage, the analysis and records were carried out at the discretion of the teachers who participated.

### 2.7. Subjects and Scenario of the Research

The Agricultural Technological Baccalaureate Center, campus 197 (CBTA No. 197), belonging to the Unit of Higher Secondary Education Agricultural Technical and Marine Sciences (UEMSTAyCM), is located in the municipality of Cajeme, Sonora, Mexico, in the so-called Yaqui Valley, cradle of the Green Revolution. On 15 February 1989, it was opened as an extension of CBTA No. 38, based in the Marte R. Gómez police station in the same municipality. In 1996, they received the category of squad. Its student population comes mainly from 16 rural communities, as well as from the municipal capital Ciudad Obregón.

The environment of the staff and the population in socioeconomic terms is medium-low. The main important economic activities are related to agriculture, livestock, and fishing, as well as the agro-industry [33].

## 3. Results and Discussion

CBTA 197 offers bivalent, schooled, and face-to-face upper secondary education. It is located in a metropolitan area of Ciudad Obregón, Sonora, Mexico. The organizational design offers a preparatory training that allows its students to enter the university and access a title of Upper Middle Technician, a title recognized by the General Directorate of Professions. The curricular plan of all the careers offered is designed by competencies; academic training has a high emphasis on practice, which is carried out in different workshops related to agricultural activities.

With regard to the workshops with teachers of the agricultural professional component, it was determined that the TAP program must be comprehensive, transmit technical knowledge, and also incorporate skills that allow the graduate to understand the environment and relate efficiently with the productive sector and its actors efficiently.

The following werediscussed in a general way, with respect to the current curriculum [34]:

Module I promote sustainable development and is taught to second-semester students who do not have the maturity and experience to organize staff for production, diagnose the agroecological environment, or plan strategies for production, as indicated by the program, so it is recommended to relocate it to the sixth semester, where they have more tools to develop the competence.

Module II uses agricultural techniques for production. The teachers agree on the general aspects of the program and the difficulty when carrying out the didactic sequences to delimit the situations raised, the priority contents are chosen, and professional competences focused on food security are suggested.

In module III, livestock species are managed, awareness is raised regarding the importance of addressing the contents from the systems approach, and food security, from the dimension of access and availability of food security.

In the case of module IV, agricultural products are processed, and it is recommended to approach it from the system approach to relate it to the environment and the previous and subsequent agricultural modules. In the same way, food security is integrated from the dimension of food use.

With respect to module V, it operates the sustainable development project. The priority contents of module I are taken up, and it promotes sustainable development. It is then incorporated and adapted, and food security is integrated from the dimension of availability.

Once the diagnosis has been made, having selected the priority contents and proposed those that complement and reinforce food security, we proceed to apply the interviews to the public and private sector, where the following results were obtained:

Eighty percent of the interviewees who work in public and private institutions argued that agricultural technicians were rated as being between regular and sufficiently trained, and that it is necessary to update the graduation profile, in the context of food security.

Regarding the question of the main limitations faced by TAP graduates, the general opinion is that graduates have a good management of the related aspects of the agricultural sector, but a more comprehensive knowledge is needed regarding the factors that impact food security in the region and in the country in order to propose strategies that correct it.

Food security is directly related to the production of food, safety, and nutrition, and to the level of excess forms of consumption of the population, respecting their culture and preferences. To achieve food security, agricultural security must be achieved; therefore, it is important to declare it in the curriculum of the TAP.

Therefore, the interviewees believe that it is necessary for the country to implement programs aimed at food security and suggest treating it in a transversal manner and

applying it as a professional component through school gardens, school feeding, and training to the educational community, because it is a way to transmit knowledge and make ties with the community.

To understand the preferences regarding the professional competence proposed for the TAP, in the consultation process, participants were provided with a survey using Google forms, with the following scale:

1 = extremely important

2 = very important

3 = quite important

4 = unimportant

In order to understand the acceptance of the proposal made to the curriculum in sixth semester students based on professional skills, it is attended with the proposal of incorporation of professional skills that allow them to perform efficiently in topics such as the pillars of food security. Regarding the assessment of the professional competences proposed by sixth semester students of the agricultural technician, the results are as follows:

Module I. Promote food security, rating 1, extremely important, 57% (65); Module II. Agricultural production system for food security, obtained a very important rating, 80% (70), and with a rating of quite important with 20% (18); Module III. Livestock production system for food security, had an extremely important rating with 83% (73), and a very important rating with 17% (15); Module IV. Agro-industrial production system for food security, obtained a rating of extremely important with 74% (65), and a valuation of very important with 17% (15), and a valuation of quite important of 9% (8); Module V. Design and execution of projects in food security, obtained an extremely important rating of 72% (63), very important with 20% (18), and finally, quite important, 8% (7).

*Proposal of Innovation of the Curriculum to Contribute to Food Security in Agricultural Technical Baccalaureate Centers*

According to [30], the career of Agricultural Technician is made up of the following Modules: (I) Promote sustainable development; (II) Employ agricultural techniques for production; (III) Manage livestock species; (IV) Process agricultural products; and (V) Operate sustainable development projects. These allow the graduate to join the labor market in the sectors of agricultural, livestock, and agro-industrial production, according to their individual needs and those of the social environment.

The study program consists of 1200 h, which are distributed in five modules, which begin in the second semester and conclude in the sixth semester. Modules I, II, and III have 272 h each, and modules IV and V are 192 h each.

It emphasizes that, according to the results of the investigation, it was determined that, in the particular case of Module I, it promotes sustainable development and its respective submodules and professional competencies. for second semester students who do not yet have the psychosocial maturity and do not reach the dimension, due to the complexity of promoting sustainable development through the sustainable management of agroecosystems that seek to have the following attributes: (a) Achieve a high level of productivity, (b) Provide reliable, stable, non-decreasing and resilient production, (c) Provide flexibility (adaptability) to adapt to new conditions of the economic and biophysical environment, (d) Fairly and equitably distribute the costs and benefits of the system, in addition to, (e) Possess an acceptable level of self-reliance (self-management) to respond to and control externally induced changes [35].

Professional competences are complemented by generic competences acquired in the professional component, complemented by those of productivity and employability, and socio-emotional skills. This is similar for the disciplinary competences that are taught in the baccalaureate. In this sense, generic competences are fundamental for acquiring the graduation profile, due to the fact that they are transversal and transferable.

Regarding the proposal generated in this research, the result obtained is presented below. The first thing to highlight is the change of name from Agricultural Technician

to Agricultural Technician in Food security. The new proposal considers the following Modules: (I) Promote food security; (II) Establish agroecological production systems; (III) Manages livestock species under safety criteria; (IV) Process traditional foods with regional identity; (V) Manage projects that contribute to food security. The proposed study program continues to be 1200 h; for the five modules the change lies in the equitable distribution of 240 h for each, Table 1.

**Table 1.** Proposal of changes in the Agricultural Technician.

| Module | Agricultural Technician | | Agricultural Technician in Food Security | |
|---|---|---|---|---|
| | Competence | Hours | Competence | Hours |
| I | Promote sustainable development | 272 | Promote food security | 240 |
| II | Employ agricultural techniques for production | 272 | Establish agroecological production systems | 240 |
| III | Manage livestock species | 272 | Manage livestock species under safety criteria | 240 |
| IV | Process agricultural products | 192 | Process traditional foods with regional identity | 240 |
| V | Operate sustainable development projects | 192 | Manage projects that contribute to food security | 240 |
| Total | | 1200 | | 1200 |

Source: Author made.

It is recommended to standardize the time of practical activities with a minimum of 40%, because of the relevance it has for technical vocational training, and 60% for conceptual theory.

Because the module can be taught by one teacher or more in parallel, or in a staggered manner depending on the number of its submodules, the conditions of the campus and the availability of teachers, a sequential distribution is suggested for a better connection, continuity, and coherence between them. In addition, professional competences must be approached from the systems approach, where each of the elements that make it up are different but are related in such a way that they are integrated into a whole [36].

On the other hand, it concludes the need to change the organization of professional competencies to sequenced content, because it allows greater orientation, delimitation, depth, congruence, and coherence for the teacher and the student, unlike with flexible organization and by submodules. Most teachers teaching the module agree that this change would improve the teaching–learning process and significantly simplify the development of didactic sequences. Likewise, the teaching of the module by a single teacher would give a better follow-up to the progress of the student with respect to the acquired competences. On the other hand, with regard to generic, productivity and employability, and socio-emotional skills, no change is proposed.

Another important point of the proposal is the articulation of professional competences with at least two of the Sustainable Development Goals of the 2030 Agenda, whose Goal 2 is "Zero hunger" and objective 4 is "To guarantee inclusive, equitable and quality education and promote lifelong learning opportunities for all", with respect to target 2.1 of the Agenda "End hunger and ensure year-round access to food for all" and target 2.4 "Ensure the sustainability of food production systems and implement resilient agricultural practices that increase productivity and production, contribute to the maintenance of ecosystems, strengthen resilience to climate change, extreme weather events, droughts, floods and other disasters, and progressively improve soil and land quality".

With regard to the quality of education, it is through the process of curricular innovation that the quality of education can be improved to promote human development, with the participation of teachers, students, and public and private institutions for a comprehensive formation, in such a way that it is recommended to carry it out periodically, due to social, technological, and scientific changes. The curriculum must keep pace with these

changes to make education relevant and of quality. In Table 2, the proposal is broken down by module, submodule, professional competence, and hours allocated.

**Table 2.** Proposal of curriculum of the agricultural technician in food security.

| Module | Submodule | Professional Competence | Hours |
|---|---|---|---|
| I. Promote food security | 1.1. Identify the dimensions of food security | 1.1.1. Discusses the right to food through literature review | 80 |
| | | 1.1.2. Recognizes the current situation of world hunger in order to measure the problem | |
| | | 1.1.3. Differentiates the dimensions of food security in order to understand its component elements | |
| | 1.2. Interprets international and national regulations | 1.2.1. Explains the objectives of sustainable development | 80 |
| | | 1.2.2. Analyzes different food programs and public policies in Mexico and in the world. | |
| | | 1.2.3. Promotes changes in public policies | |
| | 1.3. Relates environment, development, and food security | 1.3.1. Recognises the importance of biodiversity conservation in time and space | 80 |
| | | 1.3.2. Proposes improvements to rural non-farm livelihoods | |
| | | 1.3.3. Determines the importance of food in nutrition and human health | |
| II. Establish agroecological production systems | 2.1. Characterizes local agricultural production systems | 2.1.1. Learn about the history of food and agriculture | 80 |
| | | 2.1.2. Develops socio-economic and environmental diagnostics for food security | |
| | | 2.1.3. Characterizes family farming | |
| | 2.2. Establish agroecological production systems | 2.2.1. Understands agroecological principles for production | 80 |
| | | 2.2.2. Improves health conditions in communities with nutritional effects | |
| | | 2.2.3. Assess the sustainability of production systems | |
| | 2.3. Assesses the impact of climate change on food security | 2.3.1. Explain the importance of climate in production systems | 80 |
| | | 2.3.2. Investigates the importance of water availability and quality in agricultural production | |
| | | 2.3.3. Assess resilience and adaptation to climate change | |
| III. Manage livestock species under safety criteria | 3.1. Characterizes local livestock production systems | 3.1.1. Know the main characteristics of sheep and goat production systems | 80 |
| | | 3.1.2. Know the main characteristics of bovine and pig production systems | |
| | | 3.1.3. Know the main characteristics of non-traditional species production systems | |
| | 3.2. Implement good livestock production practices | 3.2.1. Analyses the principles of biosafety | 80 |
| | | 3.2.2. Recognizes the importance of traceability in livestock products | |
| | | 3.2.3. Properly manage waste generated in livestock production | |

**Table 2.** *Cont.*

| Module | Submodule | Professional Competence | Hours |
|---|---|---|---|
| | 3.3. Manages livestock production systems under safety principles | 3.3.1. Identify the importance of color, smell, taste, appearance, and nutritional value attributes | 80 |
| | | 3.3.2. Carry out the production, storage, distribution and preparation of food | |
| | | 3.3.3. Proposes pollutant reduction systems in primary production and processing units of livestock products | |
| IV. Process traditional foods with regional identity | 4.1. Identify traditional foods with regional identity in their region of residence | 4.1.1. Recognizes the importance of cultural identity and eating habits | 80 |
| | | 4.1.2. Researches the main characteristics of traditional regional cuisine | |
| | | 4.1.3. Explains what appellations of origin are in food and drink | |
| | 4.2. Characterizes local processing systems for fruit and vegetables, cereals, dairy, and meat. | 4.2.1. Defines the economic importance of local production systems | 80 |
| | | 4.2.2. Learn about the operation and production of fruit, vegetable, and cereal technologies workshop | |
| | | 4.2.3. Learn about the operation and production of the dairy and meat technologies workshop | |
| | 4.3. Proposes added value to the agricultural sector | 4.3.1. Analyses production, processing, and marketing processes | 80 |
| | | 4.3.2. Identify the importance of the circular economy and alternative markets | |
| | | 4.3.3. Proposes agri-food networks and job creation | |
| V. Manage projects that contribute to food security | 5.1. Consensus on institutional coordination actions | 5.1.1. Contributes to strengthening the local management capacity of the target population | 80 |
| | | 5.1.2. Coordinates institutional actions between officials and key entities | |
| | | 5.1.3. Manages sufficient, necessary, and current information for decision-making | |
| | 5.2. Design integral projects | 5.2.1. Promotes technical assistance and rural extension for innovation | 80 |
| | | 5.2.2. Prepares budgets and grants for strategic projects and programs | |
| | | 5.2.3. Plans and implements food security projects | |
| | 5.3. Design monitoring and evaluation plans for integral projects | 5.3.1. Identifies critical points and indicators to evaluate proposed projects | 80 |
| | | 5.3.2. Plan and implement a monitoring and evaluation plan | |
| | | 5.3.3. Systematize successful local experiences | |

Source: Author made.

Given the above, the proposal of Agricultural Technician in Food security is justified because it requires the training of technicians who play the role of managers of rural development that contribute to the achievement of food security through promoting family farming at both the local and regional levels. However, as recommended in [37], it is necessary to have a monitoring and evaluation plan that allows analyzing the labor insertion of graduates, their impact and contribution to food security, the awareness awakened about local food insecurity, and the skills developed.

These agents of change must have an in-depth knowledge of the dynamics of family farming, the interrelation between the structure of the family in rural areas, the degree of intensity of the work carried out by the family, and the technical and economic conditions in which it is carried out. Knowledge of these aspects will allow technicians to develop proposals based on technological innovation, agroecological management of natural resources, and the administration and implementation of government support programs that contribute to improving the working and living conditions of families who dedicate much of their work to agricultural activities. They will also influence the design and implementation of public policies that recognize the role and function of family farming in the strategy to achieve food security. Another element is the one that points out [38] the importance of acquiring knowledge and skills in both the handling and the preparation and consumption of certain foods and reducing the risk of contracting a disease.

The graduate will have theoretical–methodological and operational tools and will apply them in the management of food security programs and projects, based on family farming processes. In addition, they will be able to elaborate proposals that consider organizational, productive, financial, marketing, services, logistical, and regulatory aspects related to the planning, execution, and evaluation of food security projects based on the agroecological management of family farming, both locally and regionally.

## 4. Conclusions

In general terms, poverty and hunger are current problems of great importance that must be combated from all fronts, and curricular innovation in agricultural education at different educational levels is a way to address it.

In curricular innovation, teaching and the quality of the knowledge that will be transmitted are fundamental, as well as the importance of developing the skills and aptitudes necessary to meet the needs of the environment in terms of work and daily life, and that therefore is influenced by various social factors, both local and global; therefore, the proposal presented here must be adjusted to the local conditions of the campus where the career of Agricultural Technician in Food security is taught.

Regarding the originality and contributions of this work, it stands out that there are few curricula at the upper secondary level that address food security in their contents. Therefore, it is of vital importance to incorporate it, and the nature of the Agricultural Technological Baccalaureate Centers make them ideal to strengthen food security in rural areas.

Among the limitations of this study, it can be mentioned that the validation of the proposal was performed with only 10 schools, all belonging to the state of Sonora. This limits system-wide implementation.

This study aims to point out the implications of what science can contribute, from education, to overcoming the challenges faced by food security in an environment where armed conflicts, water security, pandemics, and climate change, among other global problems, put at risk the availability of food for millions of people. On the other hand, applied science must contribute to the design of solid, sustainable, and resilient agri-food systems. For future research, it is necessary to look at how to implement food security in a transversal way in the study programs of agricultural technological baccalaureate centers. Likewise, how to carry out articulation with the private and public sectors to evaluate and understand the perception of the graduate with respect to their skills and performance in promoting food security.

Finally, we consider that the ideas and results presented here can be a theoretical reference for future work on curricular innovation and food security.

**Author Contributions:** Conceptualization, I.C.C.E. and A.G.R.G.; methodology, I.C.C.E.; software, I.C.C.E.; validation, I.C.C.E. and A.G.R.G.; formal analysis, I.C.C.E. and A.G.R.G.; investigation, I.C.C.E. and A.G.R.G.; resources, I.C.C.E. and A.G.R.G.; data curation, I.C.C.E. and A.G.R.G.; writing—original draft preparation, I.C.C.E. and A.G.R.G.; writing—review and editing, I.C.C.E. and A.G.R.G.; visualization, I.C.C.E.; supervision, A.G.R.G.; project administration, I.C.C.E. and A.G.R.G.; funding acquisition, I.C.C.E. and A.G.R.G. All authors have read and agreed to the published version of the manuscript.

**Funding:** This research received no external funding.

**Institutional Review Board Statement:** The present study does not require ethical approval.

**Informed Consent Statement:** Informed consent was obtained from all subjects involved in the study.

**Data Availability Statement:** At the request of any interested party, the Excel databases used in this research can be shared by email.

**Acknowledgments:** The active participation of students, professors, directors and all those who made this research possible is acknowledged and thanked. To the Universidad Autónoma Chapingo and the Centro de Bachillerato Tecnológico Agropecuario 197, for the facilities granted for the realization of this study.

**Conflicts of Interest:** The authors declare no conflict of interest.

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
