# Peer review of "Curricular Innovation for Food Security"

_education, doi:10.3390/educsci13040374_

Round 1

Reviewer 1 Report

The title of article doesn't correspond to its content. For example, the title is Curricular Innovation for Food Safety, but author addressed the hall text to the  food security, which is different. The same thing it could be observed in Abstract, nothing mentioned about food safety.

The Introduction is too big, I recommend to reduce the information. 
Related to figure 2, it should be translate in English.  

The author practice some phrase such "in our country" - whose country?

In table 1 appearing the info about the competences, but in reality it is Module titles - it must to be corrected.

Many words in article are repeat in the same sentence.

In conclusion, the content of article should be revised and edited according to the title and take in consideration the remarks.

Author Response

ANSWERS

We are grateful for the comments and suggestions given to this work in order to improve it. For this reason, the following corrections were made

A1. An error in the translation from Spanish to English between security and safety was the cause. It has been corrected.

A2. The introduction has been modified to be more concise in the ideas presented and thus reduce it.

A3. Figure 2 was translated to English

A4. In addition to the sentence mentioned above, some other sentences where the idea was ambiguous were corrected.

A5. The module titles presented in Table 1 were drafted as professional competencies.

A6. I revise the wording of the document to avoid repetition of words.

A7. The recommendations given were carefully taken up and the document was edited to make the title consistent with the manuscript presented.

Reviewer 2 Report

Dear authors,

I appreciate the work put into the preparation of the article. However, due to the lack of a scientific approach, lack of contribution to science, lack of problems, research questions, and originality, I am forced to recommend rejection.

Improvement should focus on the case study, on the presentation of the literature context - current, similar studies, on improving originality, and contributions to science.

The article only touches on a small theme of a local nature. The article contributes nothing to science at an international level. It is a typical case study and should be defined as such.

The drawings provided are not in English.

Importantly, there is a reference to food safety in the title; meanwhile, the article is about food security.

As I mentioned, there is a lack of literature review on the topic under study.

The authors omit the FAO definition of food security. Data on world hunger should also refer to these official statistics.

The authors did not indicate the limitations of their study. They have not demonstrated the implication for science. What does the article contribute that is new? What is the element of originality? What should be future research directions?

Author Response

A1. We are grateful for the comments and suggestions given to this work in order to improve it. For this reason, the following corrections were made

A2. The document was restructured in such a way that it seeks to evidence the scientific approach of the research presented, as well as its contribution to science, the problem it addresses and its originality.

A3. There are several university programs worldwide that offer a degree in food safety. In addition, in the case of Mexico, there are more than 300 agricultural technology high schools distributed throughout the country, so that the proposal presented here can have an impact not only nationally but internationally by being a reference for bachelor's degree programs.

A4. The requested translation was done.

A5. An error in the translation, food safety was used for food safety. This situation has been resolved.

A6. The subject of study and from where it is approached is curricular innovation and education, respectively. Likewise, the topic is about an agricultural technician's curriculum, where after curricular innovation, it addresses food security. In this sense, the bibliography highlights the worldwide problem of the need to train technicians with this approach.

A7. The requested definitions were incorporated.

A8. In the conclusions section, the recommendations proposed by the reviewer are included.

Reviewer 3 Report

The aim of the paper is to propose changes for the curriculum  for higher education preparing for the profession. The author pointed out that the current methods of teaching are quite “rusty” (aren't flexible enough) and suggested not only modern and effective didactics techniques, but above all changes in the content of the curriculum aimed more at their usefulness in the profession and consistent with activities undertaken at national and international levels by various institutions and organizations. The proposed modifications in the content of teaching relate to their deeper connection with food security. This is a very important and strategic issue in the context of the problems faced by individual countries, and finally the whole world. In his work, the Author conducted research using a survey in a representative group of students. Consultations with the scientific bodies and experts were also leaded, which confirmed that changes in the curriculum at the indicated school are necessary to launch well-educated food security specialists on the market. It should be noted that the Author should use the food security concept instead of food safety in most cases in his study. The latter is narrower then the former. Therefore, the indicated change should be made in the title of the study, as well as in the text, e.g. verse 400. Besides, the  figures nr 1 and nr 2 should be prepared in the same language as the manuscript. The manuscript is clear and relevant for the presented field. The cited references concern the most recent publications. The conclusions and statements are consistent with the evidences and arguments presented and supported by the citations listed.

Author Response

The requested changes were made.  Thank you for your comments on the work presented.

Round 2

Reviewer 1 Report

I would come with some observations to the author as follows:

PARAGRAPH 52:  repeated words combination „to learn to learn”

PARAGRAPH 481:  repeated words combination „factors, local, global and global,

PARAGRAPH 222: to improve the formulas which was used to calculate the sample size, it could be used MS Office Equation tools

PARAGRAPH 409, 444: to each table the author should indicate the provenience of information related to it (Table 1. Proposal of changes in the Agricultural Technician) (Table 2. Proposal of curriculum of the agricultural technician in food security.)

Author Response

Review

I would come with some observations to the author as follows:

R1. PARAGRAPH 52:  repeated words combination „to learn to learn”

R2. PARAGRAPH 481:  repeated words combination „factors, local, global and global,”

R3. PARAGRAPH 222: to improve the formulas which was used to calculate the sample size, it could be used MS Office Equation tools

R4. PARAGRAPH 409, 444: to each table the author should indicate the provenience of information related to it (Table 1. Proposal of changes in the Agricultural Technician) (Table 2. Proposal of curriculum of the agricultural technician in food security.)

Answer to comments

We appreciate your comments and observations, we are sure they will help us to improve our work.

A1. To learn' is 'to acquire knowledge', 'to apprehend' means 'to grasp' or 'to grasp by means of the senses'. To appropriate what has been learned.

A2. The observation is correct.

A3. The comment is accepted.

A4. The origin of the information was added.

Reviewer 2 Report

 I can recommend the manuscript to be published,

Author Response

We appreciate your comments and observations, we are sure they will help us to improve our work.